# Impact of Fortified Infant Cereals on the Burden of Iron Deficiency Anemia in 6- to 23-Month-Old Indonesian Infants and Young Children: A Health Economic Simulation Model

**DOI:** 10.3390/ijerph19095416

**Published:** 2022-04-29

**Authors:** Alberto Prieto-Patron, Patrick Detzel, Rita Ramayulis, Yulianti Wibowo

**Affiliations:** 1Nestle Research, 1000 Lausanne, Switzerland; patrick.detzel@gmail.com; 2Indonesian Sport Nutritionist Association, Jakarta 10270, Indonesia; ritaramayulis03@gmail.com; 3Center for Research and Development of Public Health Efforts, Ministry of Health, Jakarta 10560, Indonesia; onkidus@gmail.com; 4Nestle Nutrition Institute Indonesia, Jakarta 12520, Indonesia; irene.gaby06@gmail.com (I.); yulianti.wibowo@id.nestle.com (Y.W.)

**Keywords:** iron deficiency anemia, fortified infant cereals, infants and young children, health economics

## Abstract

Iron deficiency and iron deficiency anemia (IDA) are highly prevalent among Indonesian infants and young children (IYC). Severe IDA hampers mental development in young children and is linked to lower quality of life and lower productivity as adults. The consumption of fortified infant cereals (FIC) increases iron intake during the weaning period, thus reducing the social burden of IDA. In this manuscript, we aimed to assess the impact of FIC on the burden of IDA on IYC in Indonesia. We analyzed data for IYC aged 6–23 months from the fifth wave (2014–2015) of the Indonesia Family Life Survey (IFLS) and the Indonesia Demographic and Health Survey 2017 (IDHS-17). We adapted a health economic simulation model to estimate the impact of FIC that accounted for lifetime health and cost consequences in terms of reduced future income and DALYs. The mean Hb level was 10.5 ± 1.4 g/dL. Consumers of FIC had a reduced burden of disease (43,000 DALYs; USD 171 million) compared with non-consumers. The consumption of fortified infant cereals plays an important role in reducing the burden of IDA, and it might complement the available strategy of nutritional interventions to address this problem in Indonesian IYC.

## 1. Introduction

In many parts of Southeast Asia, iron deficiency and iron-deficiency anemia (IDA) among infants and young children (IYC) is a widespread public health problem [1,2]. In children, IDA is associated with decreased cognitive performance and delayed motor and cognitive development; in adults, this is manifested as decreased physical performance and quality of life [3]. Current evidence indicates that IYC below 2 years of age are at very high risk of iron deficiency [4]. The South East Asian Nutrition Survey (SEANUTS) in Indonesia showed that the prevalence of anemia was around 55% in children aged 6–23 mo [5]. Despite positive trends in the reduction in micronutrient deficiencies due to nutritional programs, the percentage of children below the Indonesian nutrient intake recommendations across age groups remains high [5]. Beyond the age of 6 months, an infant’s iron needs begin to exceed that provided by breast milk. Analysis of the adjusted linear regression showed decreased iron status indicators, such as ferritin (16%) and iron (3%), after 3 to 6 mo of age [6,7]. The amount of iron provided in the breast milk may become insufficient even before 6 mo, especially when infants have suboptimal iron reserves at birth [8], which may cause an iron gap. Therefore, during the weaning period, additional sources of iron are needed in order to meet the infant’s growing iron requirements [9]. Furthermore, the complementary feeding period spans more than half of the critical first 1000 days of life [9]. Attention should therefore be paid to the type and quality of complementary foods offered to IYC during this critical stage, in order to prevent disorders associated with deficiencies in iron and other essential micronutrients [9].

A major cause of anemia among Indonesian children is low dietary iron intake [10], and intake of nutrients that support its absorption, such as vitamin C [5]. Results from the SEANUTS study in Indonesia revealed that the prevalence of anemia in IYC aged 6 to 23 mo is more than 50%; this prevalence is significantly lower for older children aged 2 to 4.9 years, which has been estimated at 11% and 17% in urban and rural areas, respectively [5]. Iron deficiency was present in 10% and 15% of 2 to 4.9 years old children in urban and rural areas, respectively. SEANUTS did not report on iron deficiency in children below the age of 2. However, in the literature, for children below the age of two, between 25% and 67% of anemia can be attributed to iron deficiency [1,11]. Additionally, a high percentage of children have dietary intakes of energy, protein, vitamin A, and vitamin C below the Indonesian recommended dietary allowance (RDA) [5]. There is evidence suggesting that complementary feeding practices in Indonesia are sub-optimal, particularly among poorer households, where IYC are fed a limited and inadequate range of complementary foods, i.e., rice-based staple foods, a low intake of animal protein, and a low intake of nutrients which support the absorption of iron [12,13]. Severe anemia may require blood transfusions, and the management of blood transfusions needs to be carefully handled to ensure safety. Hazards associated with blood transfusion and risk of allogeneic outcomes are especially of concern in children [14].

Food-based strategies have been shown to be safe and effective at addressing iron and other micronutrient deficiencies in the general population, including IYC [15]. Local regulations require that iron be added to infants’ complementary products. The minimum requirement for iron in complementary foods for infants aged 6–12 months is 3.56 mg/100 kcal [16]. Data from the FITS study indicated that iron-fortified infant cereal is a primary source of non-heme iron among U.S. infants aged 6–11.9 months [17]. These findings echo those obtained from a recent meta-analysis of 18 randomized controlled trials (including 5468 children aged 6 months to 5 years), which showed that children who consume micronutrient-fortified milk and cereal-based products have improved hemoglobin (Hb) levels (an increase of 0.87 g/dL) and a 57% lower risk of anemia compared with children who consume similar non-fortified items [18]. These findings are consistent with those obtained from other datasets around the world, including the NHANES study [19] and other randomized controlled trials [20,21] as well as those performed in developing countries such as India [22,23], Bangladesh [24], the Ivory Coast [25], Cameroon, and Vietnam [26]. Taken together, these studies suggest that fortified infant cereals are a potential source of iron for this subgroup of children in developed and developing countries.

Although fortified infant cereals are a feasible and promising vehicle for addressing iron deficiency, there is a lack of empirical data on how the consumption of fortified infant cereals may ameliorate the burden of iron deficiency in Indonesia. A nutritional assessment of complementary feeding of infants aged 6 to 12 months using the national Individual Food Consumption Survey (IFCS) in 2014 indicated that the iron, zinc, and calcium intake of Indonesian infants was mainly from manufactured complementary foods (CF), but the amount of manufactured CF was very small compared with homemade complementary foods that were largely vegetable based [27]. Understanding the importance of fortified infant cereals in reducing the risk of IDA, and working with local policy makers and healthcare practitioners to incorporate them into local complementary feeding practices is essential. The goal of this study was to explore the potential contribution of fortified infant cereals to ameliorating the burden of IDA in Indonesian children aged 6–23 months. Our analyses were based on data from the Indonesian Family Life Survey (IFLS) [28] and the Indonesia Demographic and Health Survey 2017 (IDHS-17) [29]. We estimated the current levels of Hb in Indonesian children aged 6–23 months, and from there we generated a model to evaluate the potential impact of fortified infant cereals on alleviating the burden of IDA.

## 2. Materials and Methods

### 2.1. Data Source and Study Population

Two Indonesian national data sources were combined for this study. The first data source originated from the fifth wave of the IFLS 2014–15 [28]. The second was from the IDHS-17 [9]. The IFLS is an ongoing, large-scale, longitudinal socioeconomic and health survey that has been gathering data for 21 years. The survey covers many aspects of the physical and social environment, including extensive measures of health status. Data gathered include self-reported measures of general health status, symptoms, physician-diagnosed chronic conditions, physical activities, and biomarker measurements (including but not limited to physical measurements such as height and weight, blood pressure, pulse, Hb level, and lung capacity) [8]. The IDHS-17 was carried out by the National Population and Family Planning Board (BKKBN), Statistics Indonesia (BPS), and the Ministry of Health (Kemenkes), and funded by the Indonesian government. Technical assistance was provided by the U.S. Agency for International Development [29]. The IDHS-17 is the eighth demographic survey in Indonesia conducted under the Demographic and Health Surveys (DHS) Program, with the aim of providing current data on the basic indicators of demography and health across all Indonesian provinces [29]. The rationale for using these two national surveys is the availability of variables required from both surveys for the health economic simulation, in order to meet the objectives of this study. Although the two surveys have different study designs, they were able to be combined using children’s socioeconomic strata and mother’s education levels to link percentages of fortified infant cereals consumption and hemoglobin status by SES.

The sample used for our estimation of the average Hb concentrations according to socioeconomic group consisted of 767 IYC aged 6–23 months within the IFLS, while data on the current consumption of fortified infant cereals according to socioeconomic stratification and age (in months) included 4982 IYC from the IDHS-17. The population was stratified into socioeconomic status (SES) based upon the distribution of educational attainment and wealth indices across households from the IFLS and IDHS-17 datasets, as previously done by Wieser et al. [30].

### 2.2. Estimating the Burden of IDA and the Effect of Fortified Infant Cereals

To estimate the effects of fortified infant cereals on the burden of IDA, we developed a health economic simulation model that considered lifetime health and the cost consequences, predominantly in terms of reduced future income, due to IDA in IYC aged 6–23 months. This model is similar to one developed in a previously published study performed in 6–59-month-old Pakistani infants, adapted to the context of Indonesia [29]. Figure 1 provides an overview of the main steps used in performing the calculations in this model. Further details of the model are explained in Appendix A.

As a first step, we used the delta method approximation from a regression model on Hb concentration and the share of anemia attributed to iron deficiency to estimate the prevalence of IDA across all socioeconomic groups. The prevalence of anemia was calculated based on the altitude-adjusted Hb values reported in the IFLS [10]. Then, using a systematic review of the prevalence of anemia due to iron deficiency for the South East Asia region and age group, we estimated which proportion of anemia was due to iron deficiency [11].

Next, we modeled the relationship between IDA and different scenarios with consumption of fortified infant cereals. We first projected a hypothetical Hb distribution at the population level in IYC aged 6–23 months, if fortified infant cereals were not consumed. Under this hypothetical scenario, we estimated the burden of disease using the comparative risk assessment model previously described by Wieser et al., which was used to estimate the costs of micronutrient deficiencies in Filipino children aged 6 months to 5 years [31]. We estimated the reduction in the prevalence of anemia due to consumption of fortified infant cereals by modeling the Hb trajectories and age of the children consuming fortified infant cereals, alongside their duration of consumption, compared with not consuming fortified infant cereals (Figure 2). We assumed a maximum effect after 6 months of consumption of 0.87 g/dL, as reported in a previously published systematic review by Eichler et al. [17]. The average estimates that we obtained in the 6- to 23-month-period was lower because we took into consideration a time lag between the intake of fortified infant cereals and the resulting increase in Hb.

### 2.3. Costs of the Health Consequences of IDA

We modeled the lifetime costs of IDA in our study sample of IYC aged 6–23 months who were exposed to IDA during this period. Production losses that may arise due to lowered future earnings in IYC affected by impaired cognitive or physical development were measured as future losses to gross income, based on the wages specific to their SES. Current wages according to the International Labor Organization were extrapolated to the future by assuming an income growth equal to the yearly average in the previous decade. Production losses were discounted to present values with a discount rate of 3%, and converted to U.S. dollars with the mean exchange rate of 2020. The life years lived with IDA-associated disabilities and lost due to premature death were measured in disability-adjusted life years (DALYs).

### 2.4. Sensitivity Analysis

To explore the robustness of the estimated burden, we ran the model 10,000 times to construct synthetic confidence intervals for each scenario. In Appendix B, we reported the model parameters, confidence intervals, and assumed distributions based on the reference publications. Based on the nature of the parameters, we assumed an underlying theoretical distribution. For the sensitivity analysis, we followed a similar approach to that previously published by Wieser et al. [30].

## 3. Results

The baseline population considered in the model was the 2017 cohort of babies born in Indonesia. Table 1 summarizes three basic characteristics of the population by socioeconomic decile: number of births, average hemoglobin concentration, and household income distribution. These inputs are key in the determination of the burden of IDA and the impact of fortified infant cereals on reducing the burden. Appendix B presents a list of parameters with their references and the assumed distribution for probabilistic sensitivity analysis. 

### 3.1. Baseline Hb Levels and Prevalence of IDA in the Study Sample

The estimated mean (± standard deviation) Hb level in our study sample of 6 to 23-month-old Indonesian children from the IFLS was 10.8 (±1.4) g/dL. The percentage of children with anemia was highest in the lowest SES (26.1% with mild, 38.3% with moderate, and 1.2% with severe anemia). This proportion decreased gradually from the lowest to the wealthiest households, where the percentage of children with mild, moderate, and severe anemia were 26.1%, 17.7%, and 0.1%, respectively (Figure 3). In the overall study sample, these percentages were 30.4%, 31.4%, and 1.3% with mild, moderate, and severe anemia, respectively. A total of 37% were not anemic.

### 3.2. Estimated Level and Duration of Consumption of Fortified Infant Cereals in 6 to 23-Month-Old Indonesian Children

A total of 74% of the children in our study sample consumed fortified infant cereals; 26% had never consumed them. The median duration of consumption was 5 months, with most of the subjects being short-term consumers (17% consumed fortified infant cereals for <3 months, 21% from 3–5 months). The duration of consumption was between 6–11 months for 25% of the IYC, and only 11% consumed them for 12 months or longer. When calculating the theoretical consumption of fortified infant cereals based on the recommended two servings per infant per day, we calculated a theoretical volume of approximately 30,000 tons of cereals. However, Euromonitor (a company providing aggregated sales data on a global scale) provided an estimated actual consumption volume of 15,000 tons. Altogether, these results suggest that the actual consumption of fortified infant cereals in our study sample is one serving per day. This assumption has been confirmed by internal consumer research data pointing to the dominant habit of serving fortified infant cereals as a breakfast only (not shown), and by the analysis of the IFCS 2014 showing that less than a quarter of infants aged 9 to 12 months were fed fortified infant cereals [32].

We analyzed the average Hb levels in these IYC throughout the 6 to 23-month period, according to the reported duration of consumption of fortified infant cereals (Figure 4). Hb levels increased with increasing duration of consumption of fortified infant cereals. In non-consumers (0 months; Figure 4), the average Hb levels were the lowest at 10.7 g/dL. The average levels increased to 10.8 g/dL in infants who consumed fortified infant cereals for 3 months, and to 11.15 g/dL in those who consumed fortified infant cereals for 9 months (Figure 4).

Next, we asked whether income status affected the consumption of fortified infant cereals within the 6 to 23-month period. Figure 5 depicts the percentage of children who consumed fortified infant cereals for each month, for the whole study sample as well as for the sample divided according to wealth group (low, middle, and high income). At around 6 months, less than half of the overall study sample (46%) consumed fortified infant cereals. The consumption reached a peak at 7 months (63%), with consumption levels in the overall sample dropping rapidly beyond this timepoint, to 25% at 12 months, 13% at 18 months, and 9% at 23 months (Figure 5). A higher proportion of infants in the wealthiest subgroup consumed fortified infant cereals, compared with those in the lower wealth subgroups: at 6 months, these were 39%, 49%, and 52%, dropping to 18%, 26%, and 28% at 12 months in the lowest, middle, and wealthiest subgroups, respectively. These differences between the wealth subgroups persisted up to around 20 months, but by 23 months the differences in consumption between the three wealth groups narrowed (6%, 9%, and 10% in the lowest, middle, and highest wealth subgroups, respectively).

### 3.3. Increasing the Consumption of Fortified Infant Cereals: Effects on IDA

The estimated mean (±sd) Hb level in our study sample of 6 to 23-month-old Indonesian children from the IFLS was 10.5 (±1.4) g/dL. The percentage of children with anemia was highest in the lowest SES (26.1% with mild, 38.3% with moderate, and 1.2% with severe anemia). This proportion decreased gradually from the poorest to the wealthiest households, where the percentages of children with mild, moderate, and severe anemia were 26.1%, 17.7%, and 0.1%, respectively (Figure 3). In the overall study sample, these percentages were 30.4%, 31.4%, and 1.3% with mild, moderate, and severe anemia, respectively. A total of 37% were not anemic. 

We analyzed the average Hb levels in the IYC in the 6 to 23-month period according to the reported duration of consumption of fortified infant cereals (Figure 4). Hb levels increased with increasing duration of consumption of fortified infant cereals. In non-consumers (0 months; Figure 4), the average Hb levels were the lowest at 10.7 g/dL. The average levels increased to 10.8 g/dL in infants who consumed fortified infant cereals for 3 months, and to 11.2 g/dL in those who consumed fortified infant cereals for 9 months (Figure 4).

Next, we considered whether income status affects the consumption of fortified infant cereals within the 6 to 23-month period. Figure 5 depicts the percentage of children who consumed fortified infant cereals for each month, for the whole study sample as well as for the sample divided according to wealth group (low, middle, and high income). At around 6 months, less than half of the overall study sample (46%) consumed fortified infant cereals. The consumption reached a peak at 7 months (63%), with consumption levels in the overall sample dropping rapidly beyond this timepoint, to 25% at 12 months, 13% at 18 months, and 9% at 23 months (Figure 5). A higher proportion of infants in the wealthiest subgroup consumed fortified infant cereals, compared with those in the lower wealth subgroups: at 6 months, these were 39%, 49%, and 52%, dropping to 18%, 26%, and 28% at 12 months in the lowest, middle, and wealthiest subgroups, respectively. These differences between the wealth subgroups persisted up to around 20 months, but by 23 months the differences in consumption between the three wealth groups narrowed (6%, 9%, and 10% in the lowest, middle, and highest wealth subgroups, respectively).

To estimate the impact of fortified infant cereals on reducing the burden of IDA in this population, we performed several calculations based on hypothetical scenarios of consumption of fortified infant cereals. The first possible scenario was to maintain the peak level of consumption (as seen at 7 months; Figure 5) over a duration of 3 months, i.e., to increase the duration of consumption by 2 months in those who currently consumed fortified infant cereals for only a short duration (1–2 months). The second scenario was to raise the level of consumption in the lowest wealth tertile to that of the middle tertile. 

As a first step, we calculated the current impact of consuming one serving per day of fortified infant cereals in our study sample, in terms of production losses and DALYs (Table 2). This was compared against the effect of not consuming fortified infant cereals to obtain the magnitude of the effect size. Table 2 summarizes the results of these calculations, with production losses arising from impaired physical activity, cognitive impairment, mortality, and in total, in terms of U.S. dollars and DALYs. The results showed that the consumption of fortified infant cereals yielded a reduction in production losses due to cognitive impairment and mortality (a difference of 8.5% and 13.4%, respectively, compared with not consuming fortified infant cereals; overall difference 8.5%, or USD 171 million). In terms of DALYs, the consumption of fortified infant cereals ameliorated the DALYs due to impaired physical activity, cognitive impairment, and mortality (a difference of 7.9%, 7.8%, and 12.9%, respectively, compared with non-consumers; overall difference 7.9%, or 43,000 DALYs).

The next step was to estimate the effects of increasing the duration of consumption of fortified infant cereals by 2 months in those children who were short-term consumers. This was compared against the current level of consumption, and the magnitude of the effect size was calculated in terms of U.S. dollars and DALYs (Table 2). The results indicate that increasing the duration of consumption would result in an improvement in production losses due to cognitive impairment and mortality of 1.8% and 2.8%, respectively (overall difference 1.8%, or USD 37 million). The improvements in DALYs that would result from impaired physical activity, cognitive impairment, and mortality were 1.7%, 1.7%, and 2.5%, respectively (overall difference 1.7% or 9000 DALYs; Table 3). 

We then focused on the lowest wealth tertile. We assessed the effects on the IDA burden if this group increased their level of consumption to that of the middle wealth tertile. We evaluated the effects not only of increasing the level of consumption, but also the duration of consumption, to 2 months when the duration of consumption was less than 5 months. The results from this analysis are depicted in Table 3. If consumption levels were increased to match that of the middle wealth tertile, those in the lowest tertile would experience a reduction in production losses due to cognitive impairment and mortality of 3.6% and 9.7%, respectively (overall difference 3.6% or USD 28 million). The improvement in DALYs due to impaired physical ability, cognitive impairment, and mortality would be 3.9%, 3.9%, and 10.0%, respectively (overall difference 4.0% or 10,000 DALYs; Table 4). An even larger benefit was seen if the consumption of fortified infant cereals in the lowest wealth tertile not only matched the level of those in the middle tertile, but was extended in duration by 2 months. In this case, the estimated reductions in production losses due to cognitive impairment and mortality were 4.4% and 11.5%, respectively (overall difference 4.4% or USD 33 million). The improvements in DALYs due to impaired physical ability, cognitive impairment, and mortality were 4.7%, 4.7%, and 11.8%, respectively (overall difference 4.8% or 13,000 DALYs; Table 4).

Finally, we estimated the burden of IDA if all children aged 6–23 months consumed two servings per day of fortified infant cereals, increasing the duration of consumption of fortified infant cereals by 2 months in those children who were short-term consumers (Table 4). Here, the estimated reductions in production losses due to cognitive impairment and mortality were 8.0% and 11.7%, respectively (overall difference 8.0% or USD 161 million). The improvements in DALYs due to impaired physical ability, cognitive impairment, and mortality were 7.4%, 7.4%, and 11.4%, respectively (overall difference 7.4% or 41,000 DALYs; Table 5).

### 3.4. Probabilistic Sensitivity Analysis

We performed a multivariate probabilistic sensitivity analysis (PSA) to generate a synthetic confidence interval for the baseline estimation of the current burden of IDA. We generated additional graphs, not displayed in the manuscript, for each of the scenarios considered. However, we considered that there was limited added value from the graphs as the variability was more or less in the same proportion as the base case scenario. The parameters with their assumed distributions are shown in Appendix B. Figure 6 displays the probabilistic sensitivity analysis by economic and health burden in DALYs (disability-adjusted life years) with 10,000 runs. The 95% confidence interval of the results of the model runs were between USD 0.97 and 3.64 billion of production losses and between 510 and 590 thousand estimated DALYs. We observed much wider variability in the economic losses than in the intangible cost (DALYs), as in addition to the uncertainty of the health consequences of IDA, the economic parameters play an important role. 

## 4. Discussion

Iron has been consistently identified as one of the key lacking nutrients in Indonesian infants and children [33]. A high prevalence of anemia has been reported in Indonesian children under 2 years of age [10], underscoring the importance of this subgroup as a target for interventions. Targeting children below 2 years of age, however, poses several challenges. After 6 months of age children’s nutritional requirements, alongside their limited gastric capacity, underscore the importance of providing nutrient-dense foods in addition to breast milk [34,35]. Complementary feeding is therefore of utmost importance during this critical period; any inadequacies in the nutrient content and low quality of complementary foods rapidly manifest as micronutrient deficiencies, infectious illnesses, and growth faltering [36]. A high proportion (63%) of our study population was anemic, as seen in the baseline Hb levels. The mean Hb level in our study population was 10.5 ± 1.4 g/dL, consistent with levels previously reported in the 2011 SEANUTS study [10]. Not surprisingly, the highest percentage of anemia was observed in the lowest SES group. 

A large body of research suggests that the use of fortified complementary foods, including fortified infant cereals, is a safe and effective strategy for meeting the nutritional needs of young children during this critical period [36,37]. A main goal of our study was to evaluate the current consumption of fortified infant cereals in 6 to 23-month-old Indonesian children, in order to understand the impact on Hb levels and extrapolate the effects on the burden of iron deficiency. Our findings suggested that the overall consumption of fortified infant cereals only occurs for a short period of time in some children under 2 years of age. This is consistent with consumer insights that tend to show a rather low awareness of the health benefits of infant cereals and use of the product for satiety, convenience, and safety reasons, and with the IFCS 2014. 

Next, we explored the relationship between Hb levels and the consumption of fortified infant cereals. Our results showed that Hb levels increased with increasing duration of consumption. In non-consumers, average Hb levels were the lowest at 10.71 g/dL, compared with 10.86 g/dL in infants who consumed fortified infant cereals for 3 months, and 11.15 g/dL in those who consumed fortified infant cereals for 9 months. It should be noted that our analysis did not account for other factors that may influence Hb levels, such as maternal iron status, or additional dietary components, such as animal products. Since the Hb levels in our study population also appeared to correlate with SES, it is likely that the family’s wealth status also plays a role in shaping the child’s diet (increased dietary diversity index with wealth, better access to sanitation, etc.). Nevertheless, the results from our analysis are aligned with previously published findings linking the consumption of fortified foods with higher Hb levels in children [17,26,38,39].

Considering these main findings, we evaluated several hypothetical intervention scenarios based on the consumption of fortified infant cereals and possible outcomes in terms of production losses due to IDA-related cognitive impairment and mortality. Our results showed that children who consumed one serving per day of fortified infant cereals had a reduction in future production losses due to cognitive impairment and mortality (overall difference 8.5%, or USD 171 million, equivalent to 43,000 DALYs), compared with non-consumers. If children in the lowest SES increased their level of consumption to match those in the middle wealth subgroup and prolonged their duration of consumption by 2 months, the overall difference in production losses due to cognitive impairment and mortality was 13,000 DALYs (representing a savings of USD 33 million). The largest benefits were seen if all children aged 6–23 months consumed two servings per day; in this case, the estimated reduction in production losses was 41,000 DALYs (representing a savings of USD 161 million). 

This study has three major limitations: First, there are only a few studies assessing the long-term consequences of Iron Deficiency Anemia. Although the mechanism of a lack of iron is widely understood, we acknowledge the link between iron deficiency in early childhood and health economic outcomes throughout the individual’s entire lifespan does not account for the potential influence of other factors. Second, there is an overlap between micronutrient deficiencies and health outcomes. Therefore, adding up burden of micronutrient deficiencies from individual studies may lead to an overestimation. Third, there is uncertainty as to the share of anemia due to iron deficiency, which directly affects the estimates of the model. Despite the abovementioned limitations, a health modeling approach provides a practical alternative to help guide public health actors to assess the effectiveness and cost-effectiveness of interventions. 

Improper complementary feeding practices are a common problem in developing countries. A recent cross-sectional study was performed with 392 Indonesian IYC aged 6–23 months, with the goal of analyzing complementary feeding practices and nutritional status. Results from this study indicated that complementary feeding practices were sub-optimal, particularly with respect to minimal dietary diversity, hygiene practices, and the consumption of foods rich in iron and vitamin A. Of note, the study was performed in Aceh Besar, a province with one of the highest prevalence rates of underweight and stunting in Indonesia [40]. Another study indicated that the extent of dietary inadequacy in Indonesian children varies with age, geographic region, and socioeconomic status [41]. A small cross-sectional study of 36 infants aged 7–8 months in 2018 in Indonesia indicated that IYC fed with homemade complementary foods had lower hemoglobin (10.82 g/dL + 1.20 SD) than those fed with fortified complementary foods (11.48 g/dL ± 0.85 SD) (*p* < 0.03) [42]. In a study assessing the societal costs of micronutrient deficiencies (iron, iodine, vitamin A, and zinc) in 6 to 59-month-old Pakistani children, the most severe cost consequences resulted from iron deficiency [30], underscoring the importance of iron as a target for intervention. 

In its Global Strategy for Infant and Young Child Feeding, the WHO/UNICEF recommend the use of affordable and locally available foods [43]. In an Indonesian study assessing the effectiveness of complementary feeding recommendations (CFRs) in improving maternal knowledge, feeding practices, and child intake of key nutrients, including iron, Fahmida et al. found that a major limiting factor was economic access to nutrient-dense foods [33]. Similar to many rural Indonesian families, poverty and household food insecurity were prevalent in the study population. This suggests that cost is an important factor that shapes child feeding practices, particularly among lower-income populations. 

Another concern is that the commonly used local foods are not always able to supply adequate levels of key micronutrients. In a study conducted in the Bogor Selatan subdistrict of Bogor, West Java, Indonesia, an area categorized as nutritionally vulnerable due to the high prevalence of malnutrition, survey data from three local markets indicated that theoretical iron requirements could not be achieved from local food sources. A theoretically optimized diet based on these local source foods would provide a maximum of 63% of daily iron requirements, but a worst-case scenario calculation indicated that this would supply only 26% of iron requirements. The levels of other micronutrients such as niacin, zinc, and calcium also fell short [12]. Similar nutritional inadequacies have also been found in the baseline diets of children in other developing countries [41]. Taken together, these findings suggest that for iron and other key micronutrients, complementary feeding regimens based solely on local foods may not be sufficient to ensure dietary adequacy [12], underscoring the need for alternative strategies. The fortification of widely-used complementary foods provides a means to circumvent these problems [36]. Improvements in manufacturing techniques and the local production of fortified blended cereal products have lowered costs and made these foods more affordable for low-income families [37]. Compared with other strategies, such as micronutrient supplementation, the use of food-based strategies has been shown to be more sustainable over the long term [11]. There are several advantages of fortified complementary foods such as infant cereals, including the ability to provide an appropriate balance of nutrients, the potential for minimizing microbial contamination compared with home-prepared foods, and time savings for caregivers [37]. Furthermore, the longer shelf-life of these foods renders them a practical alternative for rural families who live far from markets, and who are unable to obtain fresh nutrient-dense foods on a regular basis [33]. The Global Alliance for Improved Nutrition (GAIN) identifies fortified infant cereals as an important vehicle for combating micronutrient deficiencies in infants and young children [44]. In low-resource settings, improving the quality of complementary feeding has been ranked second only to improving the rates of exclusive breastfeeding to prevent deaths in young children [8,34,35,45]. The period of complementary feeding overlaps with at least half of the critical first 1000 days of life, a time when nutritional and biological factors play an essential role in shaping future health. The use of fortified infant cereals can be part of a comprehensive, food-based approach to address deficiencies in iron and other micronutrients in this vulnerable subgroup of children.

## 5. Conclusions

The findings from this study suggest that the consumption of fortified infant cereals (1 or 2 serving/day) could be a potential strategy for reducing the burden of IDA in Indonesian IYC aged 6–23 months, and extending the consumption period would generate considerable health and economic societal benefits. A combination of improved access to fortified infant cereals with nutrition education and nutritious homemade food might provide the most beneficial impact.

## Figures and Tables

**Figure 1 ijerph-19-05416-f001:**
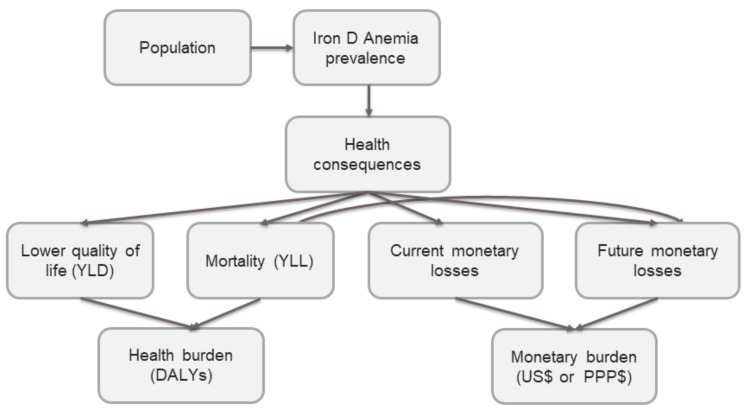
Structure of the simulation model.

**Figure 2 ijerph-19-05416-f002:**
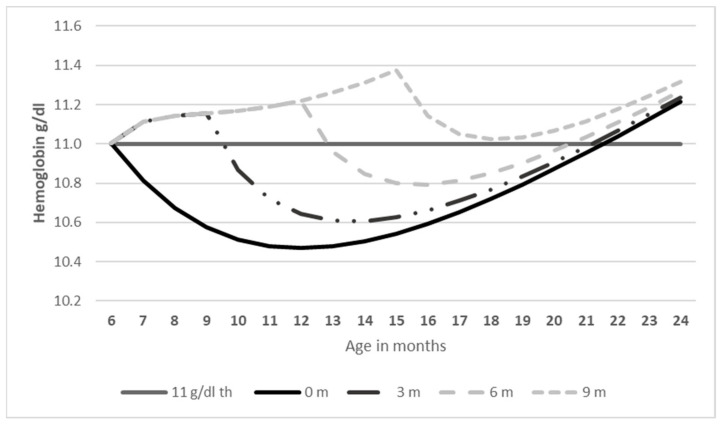
Modeling of the hemoglobin concentration trajectory according to the duration of consumption of fortified infant cereals.

**Figure 3 ijerph-19-05416-f003:**
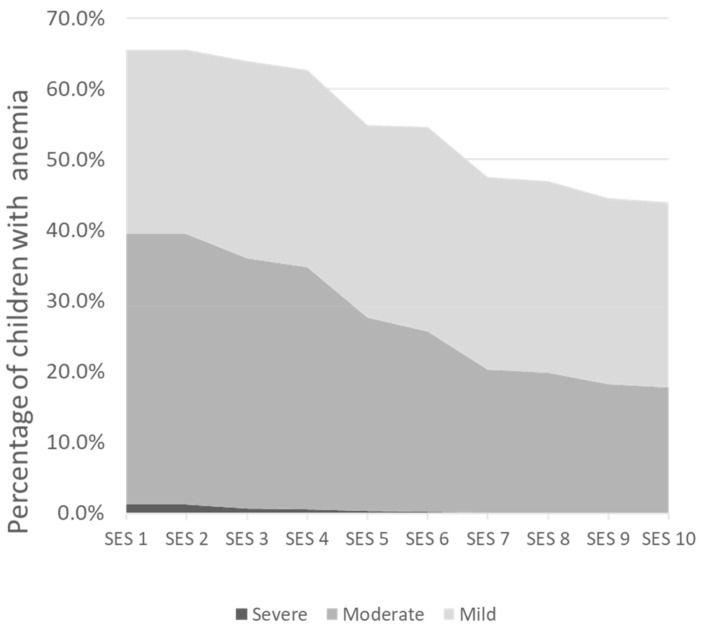
Percentage of 6 to 23-month infants and young children with anemia across the 10 socio-economic strata (SES).

**Figure 4 ijerph-19-05416-f004:**
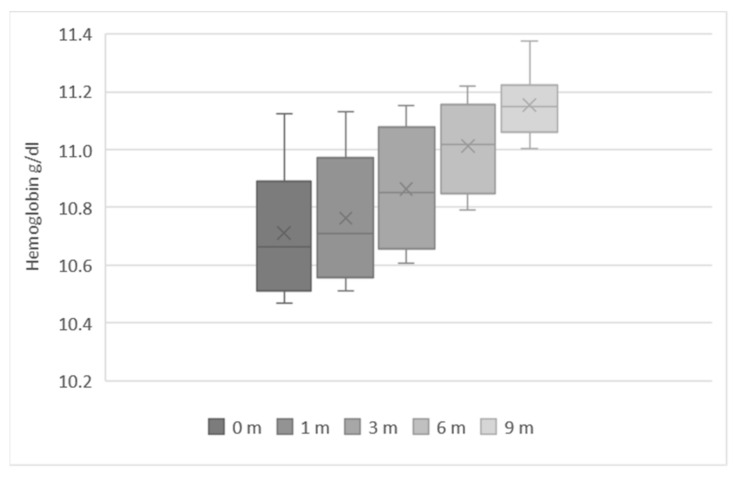
Effect of the duration of consumption of fortified infant cereals on hemoglobin levels in infants and young children aged 6–23 months.

**Figure 5 ijerph-19-05416-f005:**
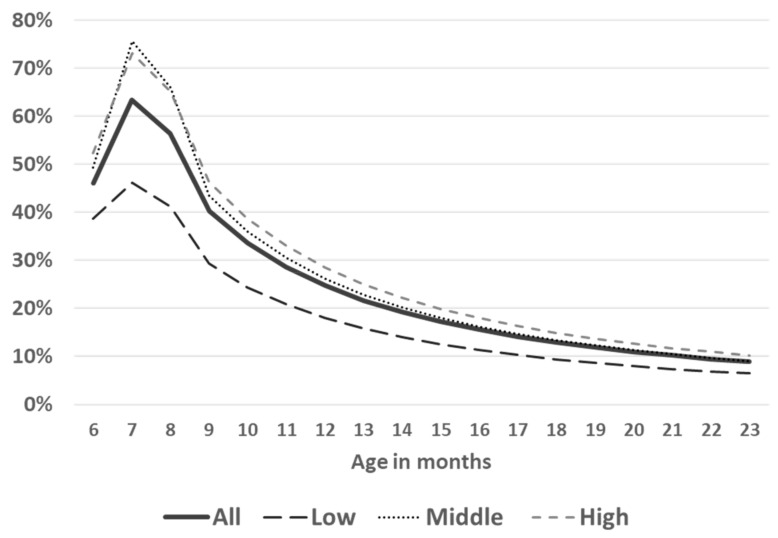
Consumption of fortified infant cereals according to wealth group.

**Figure 6 ijerph-19-05416-f006:**
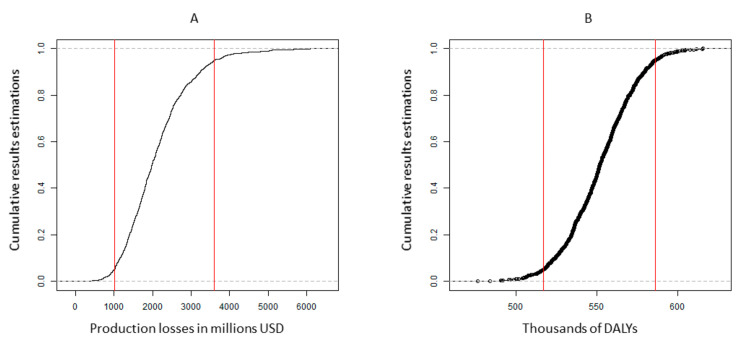
Probabilistic sensitivity analysis by economic and health burden in DALYs (disability-adjusted life years). The lines represent 95% confidence intervals.

**Table 1 ijerph-19-05416-t001:** Baseline characteristics of the population in the model by socioeconomic decile 1 to 10.

	1	2	3	4	5	6	7	8	9	10
Number of births in 2017 (in thousands) *	436	427	418	413	400	376	371	370	357	355
Average Hb in g/dL children 6 to 23 mo **	10.4	10.4	10.5	10.6	10.8	10.9	11.1	11.1	11.2	11.2
Household income distribution ***	1.92	3.21	4.23	5.18	6.28	7.6	9.29	11.6	15.9	34.8

Note: * Estimated using population weights from DHS-17. ** Average hemoglobin concentration estimated from the IFLS using a delta method approximation technique. *** Household income distribution reported by WIID, the World Income Inequality Database; UNU-WIDER.

**Table 2 ijerph-19-05416-t002:** Impact of the consumption of fortified infant cereals versus non-consumption on the burden of IDA in infants and young children aged 6–23 months.

**Production Losses (Millions of U.S. Dollars)**				
		Cognitive impairment	Mortality	Total
With current level of consumption and fortification		2005.2	4.7	2009.9
Without fortification of infant cereals		2175.3	5.3	2180.6
Difference (%)		170 (8.5%)	0.6 (13.4%)	171 (8.5%)
	**DALYs (thousands)**			
	Impaired physical activity	Cognitive impairment	Mortality	Total
With current level of consumption and fortification	51.9	495.5	5.5	552.9
Without fortification of infant cereals	56	534.1	6.2	596
Difference (%)	4 (7.9%)	39 (7.8%)	0.7 (12.9%)	43 (7.9%)

DALYs, disability-adjusted life years; IDA, iron-deficiency anemia.

**Table 3 ijerph-19-05416-t003:** Impact of increasing the duration of consumption of fortified infant cereals by 2 months on the burden of IDA in infants and young children with a short duration of consumption.

Production Losses (Millions of U.S. Dollars)				
	Cognitive impairment		Mortality	Total
Current consumption	2005.2		4.7	2009.9
Increased duration of consumption	1968.7		4.6	1973
Difference (%)	37 (1.8%)		0.1 (2.8%)	37 (1.8%)
	**DALYs (thousands)**			
	Impaired physical ability	Cognitive impairment	Mortality	Total
Current consumption	51.9	495.5	5.5	552.9
Increased duration of consumption	51	487.2	5.4	543.6
Difference (%)	0.9 (1.7%)	8.3 (1.7%)	0.1 (2.5%)	9.3 (1.7%)

DALYs, disability-adjusted life years; IDA, iron-deficiency anemia.

**Table 4 ijerph-19-05416-t004:** Reduction in the burden of IDA if the lowest wealth tertile increased consumption and duration of consumption of fortified infant cereals.

**If the Lowest Wealth Tertile Had a Similar Consumption Level as the Middle** **Tertile**				
**Production Losses (Millions of U.S. Dollars)**				
		Cognitive impairment	Mortality	Total
Current consumption		752.3	3.8	756.1
Increased consumption level		725.1	3.4	728.6
Difference (%)		27.2 (3.6%)	0.4 (9.7%)	27.6 (3.6%)
**DALYs (in thousands)**				
	Impaired physical activity	Cognitive impairment	Mortality	Total
Current consumption	23.9	231	4.7	259.6
Increased consumption level	23	222	4.2	249.2
Difference (%)	0.9 (3.9%)	9 (3.9%)	0.5 (10.0%)	10 (4.0%)
**If the lowest wealth tertile had a similar consumption level as the middle tertile and increased duration of consumption**				
**Production losses (millions of U.S. dollars)**				
		Cognitive impairment	Mortality	Total
Current consumption		752.3	3.8	756.1
Increased consumption level and duration		719.4	3.4	722.8
Difference (%)		32.9 (4.4%)	0.4 (11.5%)	33.3 (4.4%)
**DALYs (in thousands)**				
	Impaired physical ability	Cognitive impairment	Mortality	Total
Current consumption	23.9	231	4.7	259.6
Increased consumption level and duration	22.8	220.2	4.1	247.1
Difference (%)	1.1 (4.7%)	10.8 (4.7%)	0.6 (11.8%)	12.5 (4.8%)

DALYs, disability-adjusted life years; IDA, iron-deficiency anemia.

**Table 5 ijerph-19-05416-t005:** Impact on the burden of IDA if all 6–23-month infants and young children consumed two servings per day of fortified infant cereals.

Production Losses (Millions of U.S. Dollars)				
		Cognitive impairment	Mortality	Total
Current consumption		2 005.2	4.7	2 009.9
Two servings per day		1 844.5	4.1	1 848.7
Difference (%)		161 (8.0%)	1 (11.7%)	161 (8.0%)
	**DALYs (in thousands)**			
	Impaired physical ability	Cognitive impairment	Mortality	Total
Current consumption	51.9	495.5	5.5	552.9
Two servings per day	48.1	458.9	4.9	511.9
Difference (%)	3.8 (7.4%)	36.6 (7.4%)	0.6 (11.4%)	41 (7.4%)

DALYs, disability-adjusted life years; IDA, iron-deficiency anemia.

## Data Availability

Raw data for The Fifth Wave of the Indonesia Family Life Survey and the Indonesia Demographic and Health Survey 2017 are available from their respective websites: the RAND Corporation (https://www.rand.org/ (accessed on 1 March 2021)) and The Demographic Health Survey Program (https://dhsprogram.com/ (accessed on 28 February 2021)).

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
