# Peer review of "Impact of Fortified Infant Cereals on the Burden of Iron Deficiency Anemia in 6- to 23-Month-Old Indonesian Infants and Young Children: A Health Economic Simulation Model"

_ijerph, 2022, doi:10.3390/ijerph19095416_

Round 1
Reviewer 1 Report
The manuscript by Prieto-Patron et al. describes a great study and reports very interesting data. Consumption of fortified infant cereals plays an important role in reducing the burden of IDA and could be used as a potential strategy of nutritional interventions. However, some issues need to be clarified or supplemented. The comments are included below.
- Line 50-51: A major cause of anemia among Indonesian children is low dietary iron intake and nutrients which support its absorption. - What ingredients support iron absorption?
- It seems interesting to write if there are any regulations defining the permissible amount of iron supplementation in the flakes.
- Line 198-199… - Why are different units used in the text and in the figure?
- Line 180-194 and 232-246 - The same fragment of the text repeated twice ‘A total of 74% of the children in our study sample consumed fortified infant cereals; 26% had never consumed these. The median duration of consumption was 5 months, with most of the subjects being short-term consumers (17% consumed fortified infant cereals for <3 months, 21% from 3-5 months). The duration of consumption was between 6-11 months for 25% of the IYC, and only 11% consumed these for 12 months or longer. When calculating the theoretical consumption of fortified infant cereals based on the recommended 2 servings per infant per day, we calculated a theoretical volume of approximately 30,000 tons of cereals. However, Euromonitor (a company providing aggregated sales data on a global scale) provided an estimated actual consumption volume of 15,000 tons. Altogether, these results suggest that the actual consumption of fortified infant cereals in our study sample is 1 serving per day. This assumption has been confirmed by internal consumer research data pointing at the dominant habit of serving fortified infant cereals as a breakfast only (not shown) and by the analysis of the IFCS 2014 showing that less than a quarter of the infants aged 9 to 12 months were fed with fortified infant cereals.30’
- Other repetitions of the same text are also found in chapters 3.2 and 3.3.
- Line39 and 340: The same fragment of the text repeated twice.- Line39 - Beyond the age of 6 months, an infant’s iron needs begin to exceed that which is provided by breast milk; Line340 Beyond the age of 6 months, an infant’s iron needs begin to 39 exceed that is provided by breast milk.;
Author Response
Dear Reviewer,
We want to thank you for your thoughtful revision of our manuscript. Your comments and suggestions, brough us new ideas that improve the quality and easiness for the readers to go through manuscript, understand the model as well as the results and conclusions. Based on your review, we have done suggested modifications adapted the manuscript improving its quality and easiness the reader. First, we improve the introduction following your suggestions and clarifications. We check for receptions and manners to be more succinct to convey the message. We revised for consistency in the units displayed and we valued highlighting that we were reporting in different units. To better understand the context of the result we also added two appendices, one for the model further explanation and the other to clarify all the parameters used. We those modification we hope you feel that your concerns are addressed. Below we provide the detail answers to each of your concerns shared in the report.
On behalf of all authors, we want to convey our gratitude for your revision,
Alberto and Yuli.
Comments and Suggestions for Authors
The manuscript by Prieto-Patron et al. describes a great study and reports very interesting data. Consumption of fortified infant cereals plays an important role in reducing the burden of IDA and could be used as a potential strategy of nutritional interventions. However, some issues need to be clarified or supplemented. The comments are included below.
Line 50-51: A major cause of anemia among Indonesian children is low dietary iron intake and nutrients which support its absorption. - What ingredients support iron absorption?
We appreciate your question contribute to improve the introduction of the manuscript. We added in the text this information that Vitamin C is known to be the precursor of iron absorption. Based on survey done by Sandjaya et al it (SEANUTS study) was found that among children aged 0.5-1.9 years except anemia, percentage of consumption of vitamin C below Indonesian recommendation was also high (54-78%) both in urban and rural area.
It seems interesting to write if there are any regulations defining the permissible amount of iron supplementation in the flakes.
Based on Indonesian local regulation Iron is one amongst mineral that is mandatory to be added in the complementary food products for infants (6 month to 12 months). The requirement minimum amount of iron per 100kcal is 3.56mg (Indonesian FDA regulation no 24 year 2020)
Line 198-199… - Why are different units used in the text and in the figure?
Thank you for the observation, we have converted the units in g/dl also in the figure and verify consistency through the text.
Line 180-194 and 232-246 - The same fragment of the text repeated twice ‘A total of 74% of the children in our study sample consumed fortified infant cereals; 26% had never consumed these. The median duration of consumption was 5 months, with most of the subjects being short-term consumers (17% consumed fortified infant cereals for <3 months, 21% from 3-5 months). The duration of consumption was between 6-11 months for 25% of the IYC, and only 11% consumed these for 12 months or longer. When calculating the theoretical consumption of fortified infant cereals based on the recommended 2 servings per infant per day, we calculated a theoretical volume of approximately 30,000 tons of cereals. However, Euromonitor (a company providing aggregated sales data on a global scale) provided an estimated actual consumption volume of 15,000 tons. Altogether, these results suggest that the actual consumption of fortified infant cereals in our study sample is 1 serving per day. This assumption has been confirmed by internal consumer research data pointing at the dominant habit of serving fortified infant cereals as a breakfast only (not shown) and by the analysis of the IFCS 2014 showing that less than a quarter of the infants aged 9 to 12 months were fed with fortified infant cereals.30’
We appreciate the reviewer comments on the repeated paragraphs, and we changed the text accordingly.
Other repetitions of the same text are also found in chapters 3.2 and 3.3.
Line39 and 340: The same fragment of the text repeated twice.- Line39 - Beyond the age of 6 months, an infant’s iron needs begin to exceed that which is provided by breast milk; Line340 Beyond the age of 6 months, an infant’s iron needs begin to 39 exceed that is provided by breast milk.;
We appreciate the reviewer comments on the repeated paragraphs, and we changed the text accordingly.

Reviewer 2 Report
I have read the manuscript carefully, I believe the authors have done a good job and the article deserves to be published. The topic is of interest, I can suggest some improvements.
- the topic of iron deficiency anemia is a public medical problem worldwide, there is much evidence in the literature regarding the need for early and careful recognition of this disorder also as part of the international Patient Blood Management program in order to to ensure greater safety for patients and the population. This aspect could be mentioned and quoted in this document:
doi: 10.1016/ j.transci.2020.102779.
- The authors should better indicate the limitations of the study and the possibilities that the results give in terms of improvement.
Author Response
Dear Reviewer,
We want to thank you for your thoughtful revision of our manuscript. Your comments and brough us new ideas that improve the quality and easiness for the readers to go through manuscript, understand the model as well as the results and conclusions. Based on your review, we have done suggested modifications adapted the manuscript improving its quality and easiness the reader. First, we improved the context of Iron Deficiency Anemia in the context of the Patient Blood Management program. In the discussion we added a section of limitation of study which we acknowledged. Below we provide the detail answers to each of your concerns shared in the report.
On behalf of all authors, we want to convey our gratitude for your revision,
Alberto and Yuli.
Comments and Suggestions for Authors
I have read the manuscript carefully, I believe the authors have done a good job and the article deserves to be published. The topic is of interest, I can suggest some improvements.
- the topic of iron deficiency anemia is a public medical problem worldwide, there is much evidence in the literature regarding the need for early and careful recognition of this disorder also as part of the international Patient Blood Management program in order to to ensure greater safety for patients and the population. This aspect could be mentioned and quoted in this document:
doi: 10.1016/ j.transci.2020.102779.
Thank you for the suggestion. We had added reference and adapted the text accordingly in the manuscript at Line 64
- The authors should better indicate the limitations of the study and the possibilities that the results give in terms of improvement.
We added limitation part in the discussion lines 414 to 423.
Reviewer 3 Report
The authors touched on very important topics on maternal and child health. I agree with that consumption of fortified infant cereals plays an important role to alleviate anemia in the early childhood. But there are no statements about “health economic simulation model” that have been used for estimation, therefore, readers cannot reproduce the results. More concrete information on how health burdens and monetary burdens have been estimated, why simulation models adopt by authors are valid and reliable. I concluded that extensive and substantial revisions are required to satisfy the readers of IJERPH.
More concrete information is required in the methodology (line 124-136, Figure 1), such as
1) what is “population” used in the simulation model? Did authors examine entire population under 6- 23 months old?
2) how current “iron D Anemia prevalence” are defined and calculated?
3) how “health consequence”, “lower quality of life” “mortality” is estimated?
4) “monetary burden?” is per capita, per year basis? Unit of measurement is necessary.
5) descriptive statistics of the variables are missing.
I suppose authors are required to provide definition of the variables used in the simulation model, justification of using each variable, estimation procedures and equations under “health economic simulation model.” Important information to examine validity of simulation model is lacking. Readers cannot examine estimation procedures with the existing information on the manuscript.
I could not understand the scenario in line 137-151, Figure 2. How many infants aged 6- 23 months are under different four scenarios, i.e. 0m, 3m, 6m, 9m? Why every child is assumed to be same hemoglobin level (110 g/l)? Heterogeneity among children is not taken into account under the model.
No sensitivity analysis is provided. Commonly for simulation analysis, the estimation model is critically evaluated if value of important variables vary.
To examine the impact of fortified infant cereals on iron deficiency anemia, the common research procedures is conducting RCT (Randomized controlled Trial), which yield very strong evidence in the causal inference. I suggest applying RCT to reach their research goals. I suppose “health economic simulation model” advocated by authors may provide only association among iron intake and anemia prevalence even if sufficient description on the simulation model is provided. “Consumption of fortified infant cereals …as a potential strategy of nutritional interventions” (line 23-25) is misleading statement.
Author Response
Dear Reviewer,
We want to thank you for your thoughtful revision of our manuscript. Your comments, suggestions, and concerns brough us new ideas that improve the quality and easiness for the readers to go through manuscript, understand the model as well as the results and conclusions. Based on your review, we have done substantial modifications on our manuscript. Frist, we added figures, tables and modified the text accordingly to explain the model and the parametrization used to increase transparency. Second, we incorporated of a new part in the result section to discuss the probabilistic sensitivity analysis and in the appendix B we added tables with the distributions used. Third, we address concern about the validation of the model in appendix A, explaining step by step how the model is calculated and referring to previous validated models from the Institute of Health Metrix on the Global Burden of Disease approach as well as the Wieser model used in several publications. We those modification we hope you feel that your concerns are addressed. Below we provide the detail answers to each of your concerns shared in the report.
On behalf of all authors, we want to convey our gratitude for your revision,
Alberto and Yuli.
Comments and Suggestions for Authors
The authors touched on very important topics on maternal and child health. I agree with that consumption of fortified infant cereals plays an important role to alleviate anemia in the early childhood. But there are no statements about “health economic simulation model” that have been used for estimation, therefore, readers cannot reproduce the results. More concrete information on how health burdens and monetary burdens have been estimated, why simulation models adopt by authors are valid and reliable. I concluded that extensive and substantial revisions are required to satisfy the readers of IJERPH.
On behalf of all authors who have reviewed and discuss the comment we want thank and acknowledge the reviewer for his/her thoughtful analysis of the manuscript highlighting concerns that once address would improve significantly the quality of the manuscript.
More concrete information is required in the methodology (line 124-136, Figure 1), such as
We address the reviewer’s concern adding explanation of figure 1 and a section in the Appendix to show to the reader the detail calculation by example for one case scenario in one SES group.
- what is “population” used in the simulation model? Did authors examine entire population under 6- 23 months old?
- We considered one year cohort Indonesian population.
- The studied period for this population is from six to 23 months old.
- From the population surveys, we obtain data from children from six months to 23 months old children to get information about consumption of fortified baby food, hemoglobin concentration by age.
- how current “iron D Anemia prevalence” are defined and calculated?
The prevenance of Iron Deficiency Anemia (IDA) is defined as share of the anemia prevalence due to iron deficiency ad we calculated using the following steps:
- First, we estimated the Hb concentration by socio-economic decile using the delta-method approximation from a regression model on Hb concentration data from the fifth wave of fifth wave of the IFLS 2014-15.
- Then using the standard deviation on the Hb concentration, we estimated the prevalence of anemia by SES group.
- The share of anemia attributed to iron deficiency to estimate the prevalence of IDA across all socioeconomic groups from the Systematic Review of Kassebaum et al. 2014.
- how “health consequence”, “lower quality of life” “mortality” is estimated?
- The health consequences’ IDA (due to the lack of an adequate number of healthy red blood cells to carry to carry oxygen) are:
- Lower quality of life due to:
- Reduced physical activity (tiredness, physical unwellness) is a reversable condition because if the child has IDA and iron store these symptoms disappear.
- Mental impairment, suboptimal brain development due to the lack oxygen in early age, is parametrized through Lozzof study relating IDA with IQ losses. Then IQ losses in some specific cases would lead to mental impairment.
- Mortality, studies had shown higher mortality for children suffering from moderate to severe anemia. In the model this is parametrized using the population attributable fraction that considering the prevalence of IDA and child mortality. attribute a fraction of child mortality to this health condition.
- “monetary burden?” is per capita, per year basis? Unit of measurement is necessary.
- The burden of Iron Deficiency Anemia is indicated in million of US dollars per year. To clarify this point we added in appendix A, a complementary figure and further explanation how the model has been estimated.
- descriptive statistics of the variables are missing.
- We added now a descriptive on the number of births by socioeconomic group in the 2019 cohort (reference year for the model) as well as the income distribution as well as the summary table of the main parameters considered in the model.
- We added in appendix B, a summary table of all parameters used in the model with the corresponding distribution for the sensitivity analysis.
- I suppose authors are required to provide definition of the variables used in the simulation model, justification of using each variable, estimation procedures and equations under “health economic simulation model.” Important information to examine validity of simulation model is lacking. Readers cannot examine estimation procedures with the existing information on the manuscript.
- Our model derives from Wieser et al. and Plessow. Health economic model on micronutrient deficiencies. This model has been validated and published. We did minor adaptation to incorporate the dynamics on the hemoglobin path in infants six to 23 months as consumption on fortified baby food has a delayed effect on hemoglobin concentration.
- To address this concern for the readers we added the appendix A. In the appendix we explain the overall intuition on the burden of disease validated by the Institute of Health Metrics
- I could not understand the scenario in line 137-151, Figure 2. How many infants aged 6- 23 months are under different four scenarios, i.e. 0m, 3m, 6m, 9m? Why every child is assumed to be same hemoglobin level (110 g/l)? Heterogeneity among children is not taken into account under the model.
- Figure 2, it is to illustrate the typical hemoglobin path from 6 to 23 months for children consuming fortified infant cereals and usual diet without fortified food on base on the effectiveness reported in Systematic Review on clinical trials. On average, with a fortification level equivalent to the Reference Nutrient Intake (RNI) after 12 months
- With an additional calculation on the metabolism of iron absorption in children, we estimated the difference for these receiving 2 serving of fortified cereals after 9 months at 0.43 g/dl hemoglobin increase which is half of observed in the clinical trial.
- No sensitivity analysis is provided. Commonly for simulation analysis, the estimation model is critically evaluated if value of important variables vary.
- This is a very good point from the reviewer, we added a sensitivity analysis section in the results and the distribution parameters of these parameters in the appendix.
- Originally, could have added additional tables with pseudo confidence intervals generated by the model but as we have many scenarios, we consider that this would add complexity for the reader. Nevertheless, if you consider that this would add transparency we would add them into the model.
- To examine the impact of fortified infant cereals on iron deficiency anemia, the common research procedures is conducting RCT (Randomized controlled Trial), which yield very strong evidence in the causal inference. I suggest applying RCT to reach their research goals. I suppose “health economic simulation model” advocated by authors may provide only association among iron intake and anemia prevalence even if sufficient description on the simulation model is provided. “Consumption of fortified infant cereals …as a potential strategy of nutritional interventions” (line 23-25) is misleading statement.
- We fully agree that RCT are the gold standard to demonstrate causal relationship. Our study, the model uses a systematic review of RCT as main input parameter. The aim of this model is to put in perspective the learnings from systematic review of RTC in the context of Indonesian newborns cohort with respect to the strategy to cope Iron Deficiency Anemia as a public health concern. An RCT, for the spam duration that is consider in the model it would not feasible. A modeling approach provides a practical alternative to help guide public health actors to assess effectiveness and cost-effectiveness of interventions.
Round 2
Reviewer 1 Report
Most of the reviewer's remarks were taken into account in the revised text.
Please, verify one issue: Line 225: Shouldn't it be 11.15 g / dL?
Author Response
Dear Reviewer,
Thank you for your cautious reading of our manuscript and constructive comments. Your suggestions were highly appreciated. Thank you once again for your we appreciate that the error was spotted in line 225 and we corrected.
Kinds regards,
Alberto Prieto Patron & Yulianti Wibowo on behalf of all authors.
Reviewer 2 Report
The authors have greatly improved the text which I believe can be published.
Author Response
Dear Reviewer,
Thank you for your cautious reading of our manuscript and constructive comments. Your suggestions were highly appreciated. We want to show our gratitude for your endorsement and constructive comments throughout the process.
Kinds regards,
Alberto Prieto Patron & Yulianti Wibowo on behalf of all authors.
Reviewer 3 Report
I have gone through revised documents.
Figure 6 showed repetition of the runs for total production costs and DALY losses. How many runs have been done? I strongly recommend revise description of sensitivity analysis done by Plessow et al. (2015). Plessow et al. (2015) discusses number of repetition and how parameters are handled based on the predetermined distribution.
I recommend provide appendix to delineate health economic model like Wieser et al. (2013), which enables readers understand how each parameter is applied inside the simulation model.
Reference:
Wieser, S., Plessow, R., Eichler, K. et al. Burden of micronutrient deficiencies by socio-economic strata in children aged 6 months to 5 years in the Philippines. BMC Public Health 13, 1167 (2013). https://doi.org/10.1186/1471-2458-13-1167
Plessow R, Arora NK, Brunner B, Tzogiou C, Eichler K, Brügger U, et al. (2015) Social Costs of Iron Deficiency Anemia in 6–59-Month-Old Children in India. PLoS ONE 10(8): e0136581. https://doi.org/10.1371/journal.pone.0136581
Author Response
Dear Reviewer,
Thank you for your cautious reading of our manuscript and constructive comments. Your suggestions were highly appreciated, and they contributed significantly to improve the final version of the manuscript.
Following your suggestions, we:
- We improved the description of the Sensitivity Analysis by adding the explanation on how the sensitivity analysis was performed in the method section (2.4). In section (3.4) we added explanations of the PSA results specifying the number of runs (10,000), and the range on the results for the production losses and intangible cost DALYs.
- We expanded the explanation of the model in the appendix, like Wieser et al. (2013), to enable readers to understand how each parameter is applied inside the simulation model.
Additionally, as we the authors, come from different geographies and backgrounds and none of us is a native English speaker. This version will be reviewed by English Language reviewing MPDI service to ensure that the readers get through the article in a smooth and succinct form.
Kind regards,
Alberto Prieto Patron & Yulianti Wibowo on behalf of all authors.